# Phase Angle Is a Stronger Predictor of Hospital Outcome than Subjective Global Assessment—Results from the Prospective Dessau Hospital Malnutrition Study

**DOI:** 10.3390/nu14091780

**Published:** 2022-04-24

**Authors:** Mathias Plauth, Isabella Sulz, Melanie Viertel, Veronika Höfer, Mila Witt, Frank Raddatz, Michael Reich, Michael Hiesmayr, Peter Bauer

**Affiliations:** 1Department of Internal Medicine, Dessau Community General Hospital, Auenweg 38, 06847 Dessau-Rosslau, Germany; melanie.viertel@klinikum-dessau.de (M.V.); vero.hoefer@gmail.com (V.H.); milawitt@gmx.de (M.W.); raddatzfrank@yahoo.de (F.R.); michael.reich@klinikum-dessau.de (M.R.); 2Medical University Vienna, Center for Medical Statistics, Informatics and Intelligent Systems, CEMSIIS, Spitalgasse 23, 1090 Vienna, Austria; isabella.sulz@meduniwien.ac.at (I.S.); michael.hiesmayr@meduniwien.ac.at (M.H.); peter.bauer@meduniwien.ac.at (P.B.)

**Keywords:** bioimpedance analysis, screening, nutritional status, inflammatory status, medical patients, surgical patients

## Abstract

This prospective cohort study of 16,943 consecutive patients compared phase angle (PhA, foot-to-hand at 50 kHz) and subjective global assessment (SGA) to predict outcomes length of hospital stay (LOS) and in-hospital mortality in patients at risk of malnutrition (NRS-2002 ≥ 3). In 1505 patients, the independent effects on LOS were determined by competing risk analysis and on mortality by logistic regression. In model I, including influence factors age, sex, BMI, and diagnoses, malnourished (SGA B and C) patients had a lower chance for a regular discharge (HR 0.74; 95%CI 0.69–0.79) and an increased risk of mortality (OR 2.87; 95%CI 1.38–5.94). The association of SGA and outcomes regular discharge and mortality was completely abrogated when PhA was added (model II). Low PhA reduced the chance of a regular discharge by 53% in patients with a PhA ≤ 3° (HR 0.47; 95%CI 0.39–0.56) as compared to PhA > 5°. Mortality was reduced by 56% for each 1° of PhA (OR 0.44; 95%CI 0.32–0.61). Even when CRP was added in model III, PhA ≤ 3° was associated with a 41% lower chance for a regular discharge (HR 0.59; 95%CI 0.48–0.72). In patients at risk of malnutrition, the objective measure PhA was a stronger predictor of LOS and mortality than SGA.

## 1. Introduction

Hospital malnutrition is a common problem worldwide and has been shown to be associated with increased mortality and prolonged length of stay (LOS), requiring increased resource allocation [1,2,3,4,5,6,7]. To diagnose malnutrition, a number of nutrition indicators have been used for assessing the various dimensions of nutritional status: indicators of impaired nutrient balance, such as reduced food intake and non-volitional weight loss [8]; anthropometric measures, such as BMI, midarm circumference and triceps skinfolds, skeletal muscle mass [9,10]; biochemical analysis, such as albumin and/or other serum proteins [11]; and functional measures such as handgrip strength and/or other tests of muscular function [12,13]. Moreover, in the acute care setting, nutritional status and disease-driven effectors such as inflammatory status determine the metabolic challenge the sick patient is facing. As a consequence, multi-component assessment tools have been devised with the aim of predicting nutrition-related outcomes in hospitalized patients. SGA, a composite evaluation of nutritional status at the bedside, was shown to have better sensitivity and specificity than individual nutrition indicators [14], has since been implemented by researchers studying malnutrition [1,2,15,16], and is still regarded as a “gold standard”. Nevertheless, the SGA has been criticized repeatedly for its subjectiveness, and a number of composite tools including objective measures have been proposed to better predict outcome [17,18,19,20,21].

To avoid subjectiveness inherent in narrative criteria, objective, easy-to-perform, valid, inexpensive, and robust diagnostic criteria are needed that can be implemented by physicians and other medical staff without special training in clinical nutrition and without the need for sophisticated interpretation. Since malnutrition is characterized by impaired body composition and tissue function [22], it appears logical to base its diagnosis on a measure of the prevailing impairment of body composition and tissue function, such as phase angle (PhA). PhA is calculated from the arctangent of resistance and reactance of an alternative current as it passes through the body in the process of bioelectrical impedance analysis (BIA) [23,24]. Resistance provides an estimate of extracellular fluid, while reactance is affected by type and mass of body cells and their membrane integrity [24]; therefore, PhA can be viewed as an integral measure of body composition and tissue function [23,24,25]. Age, sex, and BMI are the main constitutional factors affecting PhA, which is also modified by level of physical activity and inflammation [26,27,28,29]. Phase-sensitive monofrequency bioimpedance analysis (BIA) measurements at 50 kHz provide resistance, reactance, and PhA as crude readouts, obviating potential error due to regression equations for estimation of body compartments [24,30]. Low PhA has been shown to be associated with sarcopenia [31], frailty [32], prolonged length of hospital stay [33,34,35,36,37], and mortality in various conditions [25,38,39,40].

We therefore hypothesized that PhA as an integral measure of nutritional metabolic status, with its components body cell mass and function and hydration status [23,24], could serve as an objective assessment tool instead of composite and narrative assessment tools in hospitalized patients found to be at malnutrition risk after prior screening (NRS-2002 score ≥ 3). The aim of the study was to test, in a prospective head-to-head comparison, the performance of PhA and SGA in the prediction of outcomes such as time until regular discharge and in-hospital mortality in patients at risk for malnutrition in medical and surgical specialties of a tertiary care community hospital.

## 2. Patients and Methods

### 2.1. Patients

All adult patients admitted to Dessau Community Hospital as in-patients were screened for malnutrition risk, and those with an NRS ≥3 were eligible to participate in this prospective cohort study with the following exceptions: admission to intensive care unit, obstetric ward, or ophthalmology ward. Only patients who gave written informed consent on admission were enrolled in the study. Recruitment had to be stopped prematurely due to the COVID-19 pandemic, thus limiting the number of patients admitted to 16,943. For primary analysis, all admissions were included, and for sensitivity analysis, any readmission was excluded. The study protocol was approved by the Ethics Committee of local authorities (Ärztekammer Sachsen-Anhalt), and the study was registered (DRKS00025307).

### 2.2. Estimation of Sample Size

Malnutrition risk screening has been a mandatory procedure of the admission process in all departments of Dessau Community Hospital since 2014 using NRS-2002 on admission and every week thereafter as needed [41]. A pilot analysis of 8267 patients admitted to 15 out of 18 wards within a 6-month period showed a screening rate of 90% identifying 22% of patients at risk for malnutrition on the basis of an NRS-2002 score ≥ 3. As staff of the study team would be available on working days only, we calculated that there would be no data in up to 10% of patients with a very short LOS either due to death very early after admission or discharge too early for the study staff to capture all data. The latter would apply to patients admitted between Friday afternoon and Monday morning or during Bank Holidays. Ultimately, we calculated that overall ~20,000 patients would be admitted to 18 wards within a 12-month period, yielding ~4000 patients at risk of malnutrition.

### 2.3. Data Collection

After in-depth training regarding all techniques used, members of the study nutrition team were allocated to individual wards for tracking newly admitted patients, providing study information to patients, performing interviews, obtaining all measurements including BIA in a standardized procedure, and entering all data into the electronic patient chart at bedside after written informed consent was obtained. The demographic profile of the study sample included the following variables: age [years], height [cm], body weight [kg], sex [M/F], ICD-10 code of main diagnosis at discharge as assigned by professional coding staff according to current coding guidelines. Risk factors on a continuous scale were categorized into 4 categories using quartiles that were adapted to common cut-offs as specifically indicated for each variable. For weight loss, a fifth category “no weight loss” was used when the variable weight loss in the database was zero.

For each patient, hospital LOS was extracted from the electronic patient chart defining day of admission as day one and the day of discharge as the final day. When patients were transferred between wards or departments during one treatment case, this was considered as one case, and total LOS was extracted and entered into the study file. Outcome variables LOS [days], regular discharge defined as discharge home by the caring physician, and in-hospital mortality as well as readmission rate were extracted from the electronic patient chart and entered into the study file, making data available in digital format for later analysis. Follow-up was complete for all patients.

### 2.4. Malnutrition Risk Screening

Screening was done by nursing staff; all nursing staff had received appropriate training for implementing NRS-2002. Results were entered into the appropriate form of the electronic patient record at bedside.

### 2.5. Subjective Global Assessment (SGA)

SGA [14] was performed by study nutrition team staff, and data were entered into the appropriate form in the electronic patient record at bedside. Patients were categorized as not malnourished (SGA A), moderately malnourished (SGA B), or severely malnourished (SGA C). Patients of SGA categories B and C were considered malnourished [1,2,16], and thus SGA B and C vs. A was used in multivariate analyses. SGA was always done before obtaining BIA measurements in order to exclude potential bias.

### 2.6. Phase Angle (PhA)

BIA was performed by study nutrition team staff on the day of screening or the next working day. All BIA measurements were obtained in a standardized fashion [30] with patients in the supine position as foot-to-hand measurement in the morning. After skin preparation, electrodes (BIANOSTIC AT-electrodes; Data Input GmbH, Darmstadt, Germany) were placed on the dominant side. All measurements were performed with a phase-sensitive monofrequency device (NutriBox, Data Input GmbH, Darmstadt, Germany) applying a current of 0.8 mA at 50 kHz as used in the reference value material published by Bosy-Westphal [26]. BIA raw data resistance (precision ± 1%), reactance (precision ± 2.5%), and PhA were entered into the electronic patient chart on the day of the measurement. On this occasion, PhA was classified as either below or ≥5th percentile [26], and this was entered into the patient chart as well.

### 2.7. PANDORA-Score

All seven items (age, BMI, self-reported mobility, self-reported food consumption, medical specialty, cancer, fluid status) of the PANDORA score [42] were obtained by study nutrition team staff and entered into the appropriate form of the electronic patient chart at bedside. In multivariate modeling, PANDORA variables were used individually with the same categories as in the score. On occasion of this interview, study nutrition team staff also obtained data on the extent [kg] and extent over time [kg/month] of patient reported weight loss, which was classified as follows: weight gain, no weight change, weight loss up to 5%, weight loss 5–10%, and weight loss >10%.

### 2.8. Inflammatory Status

CRP was extracted from the electronic patient chart for assessing inflammatory status [43] in 4 categories: < 10, 10–100, > 100 mg/L, and missing.

### 2.9. Statistical Analysis

Data are given as counts with percentage or median with interquartile range (IQR) as appropriate or mean with standard deviation (SD). LOS for all outcomes was compared with Wilcoxon U test between groups. Cumulative incidence curves were plotted for regular discharge and death for SGA categories A/B/C and for PhA numerical categories (PhA ≤ 3.0°, >3.0–4.0°, >4.0–5.0°, >5.0°) as well as PhA classified as either below the 5th percentile or ≥ 5th percentile according to Bosy-Westphal [26]. Unless indicated otherwise, the time until regular discharge is referred to as LOS hereafter.

For the outcome regular discharge, there was a sufficiently large number of events to apply a multivariate Fine and Gray proportional hazard model for competing risks with admitting wards as clusters. For the outcome in-hospital mortality, a corresponding logistic regression model was applied with a reduced set of influence variables owing to the limited number of events. In addition, a ROC-curve for the outcome in-hospital mortality versus PhA was calculated. Three sensitivity analyses were performed: (i) The main analysis was repeated with COX cause-specific proportional hazard modeling to check stability of the results for different types of analysis. (ii) Fine and Gray analysis was applied to first admissions only. (iii) Additional risk factors from the PANDORA score, a measure of mortality risk, and weight loss were added to the model of the main analysis (online Appendix A). Goodness of fit was determined by using concordance. Computations were done with R 4.1.1 with the survival, survminer, ggplot2, geepack and pROC packages.

## 3. Results

Between May 2019 and March 2020, 16,943 adults were admitted as in-patients, of whom 14,150 (84%) were screened for malnutrition risk using NRS-2002. Recruitment was interrupted during Christmas week due to scheduled lockdown of many wards. Furthermore, recruitment had to be stopped 9 weeks short of schedule due to hospital hygiene measures taken in order to contain COVID-19.

Screening identified 2695 (19%) patients at risk (NRS score ≥ 3). Of those, 1505 patients were enrolled, while 1190 could not be enrolled for reasons such as severe physical and/or cognitive impairment precluding informed consent (*n* = 284); discharge (*n* = 386), transfer (*n* = 26), or death (*n* = 45) prior to contact with the study nutrition team (all three *n* = 457); conditions precluding BIA measurements such as active electronic medical devices, limb amputations, skin lesions, oedema, or inability to assume the standardized prone position (*n* = 254); or other (*n* = 105). A total of 90 refused to consent (Figure 1).

A total of 61% of patients at risk for malnutrition were classified malnourished in terms of SGA B and C (n = 914), and 39% were well nourished (SGA A, n = 584). Almost half of the patients experienced weight loss ≥ 5% before admission (Table 1). Out of all patients, 52% had a phase angle below the 5th percentile (Table 2). Neoplasms (24%) constituted the largest disease group, with digestive (19%) and cardiovascular (12%) disorders placing second and third (Table 3). A total of 1346 (89.4%) patients achieved a regular discharge, 50 (3.3%) patients died in hospital, and 109 (7.3%) had other outcomes such as being discharged to another institution or unscheduled discharge. Median hospital stay was 8 days (IQR 5–15) for all outcomes.

To analyze the performance of SGA or PhA for the outcome regular discharge, we performed multivariate analyses of competing risks using three proportional hazards models (Table 4). In model I, we found the hazard for being discharged regularly (to a good approximation, being the chance for a regular discharge during a time unit of one day) to be reduced by 26% in SGA B and C patients when compared to SGA A (HR 0.74; CI 0.69–0.79; *p* < 0.0001). The association of SGA category and patient outcome in terms of LOS or death is clearly visualized by cumulative incidence curves (Figure 2) and shows that SGA B and SGA C have a similar association with time to discharge but a highly different association with death. The same phenomenon can be observed for the association of the four PhA numerical categories and the outcomes LOS or death (Figure 3).

The association of SGA and outcome regular discharge was abrogated when PhA was added in model II (HR 0.94; CI 0.85–1.05, *p* = 0.27), whereas PhA decreased the chance for a regular discharge for each 1° by 21%, 13%, and 19% between four stepwise-decreasing PhA categories (Table 4). The chance of regular discharge was reduced by 53% for patients with PhA ≤ 3° compared with PhA > 5° (HR 0.47; CI 0.39–0.56, *p* < 0.0001).

Furthermore, when CRP and PhA were included in model III, SGA B and C was no longer associated with the outcome regular discharge (HR 0.98; CI 0.87–1.1, *p* = 0.7) (Table 4). Low PhA, however, remained associated with a lower chance for a regular discharge by up to 41% in patients with a PhA ≤ 3° (HR 0.59; CI 0.48–0.72, *p* < 0.0001). Inflammatory status was associated with a progressive decrease in the chance for a regular discharge by 25% for CRP 10–100 mg/L (HR 0.75 CI 0.63–0.88, *p* < 0.001) and 46% for CRP > 100 mg/L (HR 0.54, CI 0.44–0.65, *p* < 0.0001). It should be noted, however, that the subgroup of patients (13%) without CRP measurement had a much higher chance for earlier discharge (HR 1.65, CI 1.27–2.16, *p* < 0.001). CRP measurements could not be ordered by the study nutrition team but only by the physician in charge as deemed indicated in the individual patient. The concordance index as an indicator for the model fit was C = 0.60 for model I, C = 0.63 for model II, and C = 0.66 for model III, indicating that there are limitations when trying to predict the chance of discharge from variables at hospital admission.

Characterizing disease burden by disease entity according to ICD-10 category, we also found significant associations. Using digestive disorders as a reference, neoplastic, circulatory, respiratory, and musculoskeletal diseases as well as injury or poisoning were associated with a significantly lower chance for regular discharge regardless of the model used in the competing risk analysis (Table 4). In model II and III, age ≤65 was associated with a lower chance for a regular discharge.

In the three sensitivity analyses, we found that using COX modeling instead of Fine and Gray confirmed the loss of association of SGA B and C (vs. A as reference) after inclusion of PhA as well as PhA and CRP (Appendix A). Using only first admissions and thus excluding all readmitted patients in the Fine and Gray model did not indicate any major effects on estimates. Only some ICD-10 categories became additionally significant (Appendix A). The third sensitivity analysis including items of the PANDORA score showed that patients with severely impaired self-reported mobility (HR 0.68; 0.55–0.86; *p* < 0.001) or reduced food consumption (HR 0.83; 0.72–0.97; *p* < 0.05) had a lower chance for a regular discharge (Appendix A); neither fluid status nor weight loss were associated with outcome. This sensitivity analysis again showed the loss of association between SGA B and C (vs. A as reference) and regular discharge when PhA was included in the model. Furthermore, the association between PhA and a regular discharge was confirmed but at a weaker level. Patients with a PhA ≤ 3° had a 28% (HR 0.72, CI 0.57–0.91, *p* < 0.01 vs. PhA > 5°) lower chance for a regular discharge when CRP, low mobility, weight loss, and amount eaten were added to the model (Appendix A).

For the outcome in-hospital mortality, we applied two simplified models of logistic regression with a reduced set of regressor variables owing to the limited number of events. Mortality was reduced by 56% for each 1° of PhA (OR 0.44; 95%CI 0.32–0.61). Again, the superiority of PhA over SGA was confirmed (Table 5): SGA had no influence on in-hospital mortality once PhA was included in the model. The predictive power of PhA is underscored by the resulting ROC-curve for the outcome in-hospital mortality, with an area under the curve of 0.7 (95%CI 0.633–0.767; Figure 4). The power of PhA in predicting in-hospital mortality is even more striking when patients were classified according to a PhA either <5th percentile (*n* = 777) or ≥5th percentile (*n* = 565) showing large differences between the cumulative incidence curves, with the disjoint pointwise 95%CIs indicating clear statistical significance (Figure 5A). Likewise, patients with a PhA < 5th percentile had a significantly lower chance for a regular discharge, as shown by the clearly separated cumulative incidence curves (Figure 5B).

## 4. Discussion

This prospective cohort study analyzing PhA and SGA in a head-to-head comparison shows that PhA is a stronger predictor than SGA of outcomes length of stay and mortality in hospitalized patients at risk of malnutrition. Much of current knowledge on the epidemiology and the unfavorable risk profile of hospital malnutrition is based on research using SGA as the diagnostic “gold standard”. SGA is a composite encompassing symptoms of body weight change, food intake, gastrointestinal nutrition impact symptoms, and general physical functioning as well as an assessment of disease-associated metabolic stress, fat stores, muscle loss, and hydration status [14]. Multivariable analyses of our data confirm the ability of SGA to predict hospital outcome, showing that malnutrition (SGA B and C) was associated with a 26% lower chance for a regular discharge and a 2.7-fold increase in mortality risk. The predictive power of SGA, however, was abrogated when PhA was included in our multivariable analyses. Thus, our findings not only confirm previous observations showing an association of low PhA [33,34,35,36,37] or low BIA-derived FFMI [44] with prolonged LOS and mortality [25,38,39,40] but suggest that, in a head-to-head comparison with SGA, PhA is a superior tool for the assessment of patients’ metabolic resources, which traditionally have been addressed as nutritional status.

Our findings raise a number of issues. The use of PhA as an indicator of nutritional status has been criticized due to the fact that PhA, in addition to cell mass and cell integrity, also reflects hydration status. Hydration status is one item of the PANDORA score and was documented prospectively in all patients but showed no association with outcome in a sensitivity analysis. Loss of intracellular water and expansion of extracellular water, including the occurrence of edema and anasarca, are integral and long-known features of malnutrition [45,46]. Successful nutrition therapy is accompanied by elimination of fluid excess [47]. Moreover, the assessment of hydration status is an integral component of SGA itself [14]. Therefore, it is questionable to consider fluid status a confounder rather than recognize fluid status an integral component of metabolic nutritional status. Thus, by providing an integral measure of body cell mass and integrity as well as hydration status, PhA offers the opportunity to serve as an integral indicator of patients’ metabolic resources, i.e., nutritional status.

The use of PhA as a crude BIA readout instead of FFMI or other BIA-derived markers of nutritional status offers the advantage that derived variables may be flawed due to numerous inherent assumptions which may be erroneous in severely ill patients unless validated for this particular condition [23,24]. Moreover, total body water by definition is a component of fat-free mass and thereby of FFMI as well. Furthermore, FFMI or other BIA derived variables are not independent from the influence of hydration status inherent in the BIA raw data. Finally, PhA as a crude BIA readout is available free and independent from manufacturer-owned algorithms [48].

The majority of our patient cohort (Table 2) had a PhA < 5th percentile of a large reference population [26]. This high proportion is not surprising, considering that this was a cohort of patients at risk of malnutrition selected by systematic screening. In good agreement with other reports [25,27,39,49], in our cohort a PhA < 5th percentile was associated with a significantly lower chance for regular discharge as well as a higher risk of in-hospital mortality. The strong impact of PhA on outcome is also exemplified by our observation that in patients with a numerical PhA ≤ 3°, the chance for a regular discharge was reduced by 53%, as compared to 26% for malnourished patients (SGA B and C). The association of low PhA and prolonged LOS has been observed previously, but a direct comparison between PhA and SGA has not been reported [33,34,35,36,37]. Moreover, such observations raise the question of which cut-off is appropriate to classify low PhA as opposed to normal PhA [27]. Using 5.0° (M) and 4.6° (F) as cut-offs, Kyle et al. reported low PhA of 8.3% and 29.8% in patient cohorts with a substantial prevalence of malnutrition (SGA B and C 49.9% and 61.5%) [33,50]. Using different reference values, however, other investigators found a low PhA in 56.5% or 57.5% of cohorts with a comparable proportion of malnourished patients (54.5% and 57.5%) [51,52]. Obviously, there is only partial agreement between PhA measured by different devices and SGA obtained in different modifications as indicators of nutritional metabolic status [53]. Therefore, the use of an appropriate reference pertinent to ethnicity and global region of the study sample [26,38,54] seems of utmost importance. Therefore, in our study, PhA was classified as < 5th percentile or above using age, BMI, and sex-specific reference values obtained from more than 200,000 individuals in Germany [26]. Furthermore, since instrumental sensitivity may differ between BIA devices, we chose a device which was used for the large German reference sample [26] and in studies showing PhA as an independent predictor of physical function and clinical outcome [27,49].

As far as nutrition is concerned, hospital outcome is determined by disease-related variables such as inflammation on the one hand [43] and the patients’ metabolic resources on the other. Impaired resources as assessed by low handgrip strength have been shown to predict not only mortality but also a favorable response to nutrition support [55]. Interestingly, PhA can detect sarcopenia and low muscle quality with good accuracy [31,56], and thus, PhA may be used as a proxy when measurement of hand grip strength is not available. Frailty and sarcopenia have been shown to be associated with unfavorable outcome in many conditions [13] and using low PhA as a proxy for complex assessment of frailty has been suggested [32]. In our cohort, however, sensitivity analysis showed an independent association of severely impaired self-reported mobility with a lower chance for a regular discharge. In good agreement with the NutritionDay^®^ findings [8], low food consumption was associated with poor outcome in our cohort. Disease entity was a strong predictor of outcome, showing a significant reduction by up to 48% of the chance for a regular discharge in patients suffering from neoplastic, respiratory, or circulatory diseases. Taking this disease spectrum into consideration, it is not surprising to find a lower chance for a regular discharge in the age group ≤ 65 yr. The magnitude of inflammatory drive, too, was associated with a reduction in the chance for a regular discharge (model III). This finding adds to the powerful role of inflammatory status as a driver of metabolism observed in the EFFORT trial, showing no response to nutritional intervention in patients with a high inflammatory drive [43]. Recently, investigators proposed PhA as a proxy for the assessment of meta-inflammation [29]. In our cohort, introducing inflammatory status in model III did not invalidate PhA as a strong independent predictor of outcome. This observation adds further evidence to understanding PhA as a powerful objective measure of nutritional metabolic state as an indicator of the patients’ metabolic resources for predicting clinical outcome.

### Limitations and Strengths

There are limitations to our study recruiting patients from a single institution only, and therefore, our findings and conclusions may not be generalizable to other patient cohorts. Due to the limited number of events, the head-to-head comparison between PhA and SGA for their association with mortality could only be performed in a simplified model. In our cohort, we observed a lower prevalence of malnutrition risk (19%) than the 28% reported in the pilot study of the EFFORT trial [57], which may be due to the high screening rate (84%) in our study and the fact that our cohort comprised a large patient spectrum of medical and surgical specialties, while in the Swiss cohort, only patients from medical wards were included. Therefore, it is not surprising that in our cohort, neoplastic (24%), digestive (19%), and cardiovascular disease (12%) were the three largest ICD-10 groups, while in the Swiss studies, infectious (18% and 30%), cardiovascular (27% and 10%), and neoplastic (13% and 19%) disease ranked at the top [43,57]. Our study may be criticized for focusing on a patient cohort selected by prior nutrition risk screening. Such a sequential approach, however, of assessing nutritional status as a second step after prior systematic screening thoughtfully has been proposed by the Global Leadership Initiative on Malnutrition (GLIM) and ESPEN [21,58].

Despite such limitations, we consider our findings valuable and relevant to clinical nutrition practice. By design, our study provides real-world data from patients of a large spectrum of medical and surgical specialties of an acute care tertiary institution practicing systematic nutrition risk screening independent from the current study already for 4 years. Moreover, the prospective design of the study and the systematic feeding of source data into the electronic patient chart at bedside ensured high data quality and a low number of missing values.

Malnutrition is a clinically relevant nutritional metabolic risk. Currently, its diagnosis requires meeting one of several composite criteria attempting to provide a measure for the ultimately narrative concept of malnutrition [21,22,58]. Our findings, however, show that in patients at risk of malnutrition, a PhA < 5th percentile provides a single objective and robust non-invasive measure, obviating the need for composite tools for the assessment of nutritional metabolic status. Understanding nutritional status as an indicator of the nutritional component of the patients’ overall metabolic status and, as a consequence, malnutrition as a condition with an increased nutritional metabolic risk, our data show that PhA is a much stronger indicator of an increased nutritional metabolic risk than SGA. Another strength is that the result of PhA outperforming the human-resource-intensive SGA evaluation in the association with the outcome regular discharge could be confirmed in the association with the outcome in-hospital death. The robustness of our results is further supported by the similarity of results obtained with different statistical models in the sensitivity analyses.

## 5. Conclusions

In a mixed cohort of patients from a large spectrum of medical and surgical specialties, we found PhA to be a stronger predictor than SGA of clinical outcomes length of stay and mortality, even when powerful confounders such as inflammatory drive were taken into account. Malnutrition risk screening followed by the determination of PhA in those at risk is a low-cost, non-invasive, and objective procedure providing a single numerical readout without the need for sophisticated interpretation. We therefore propose to use PhA < 5th percentile of appropriate reference values as an objective indicator of impaired nutritional metabolic state, i.e., malnutrition. Future studies should address the potential of PhA to identify those who will benefit from nutrition support.

## Figures and Tables

**Figure 1 nutrients-14-01780-f001:**
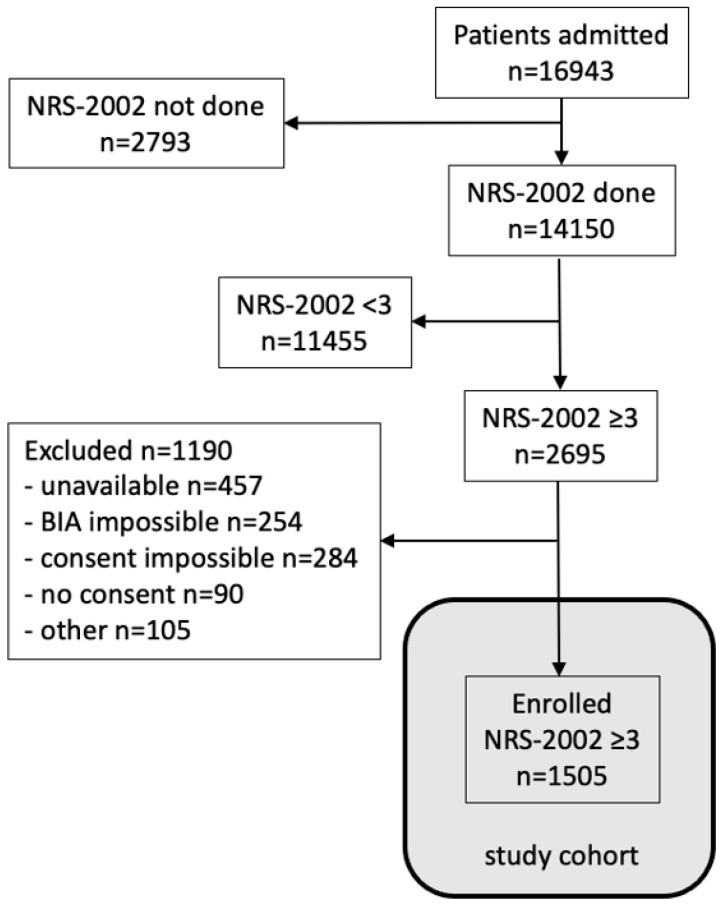
Flow diagram of study population for cohort 0 (all patients screened) and study cohort (consenting patients screened NRS ≥ 3). The multivariate analysis was performed on the study cohort.

**Figure 2 nutrients-14-01780-f002:**
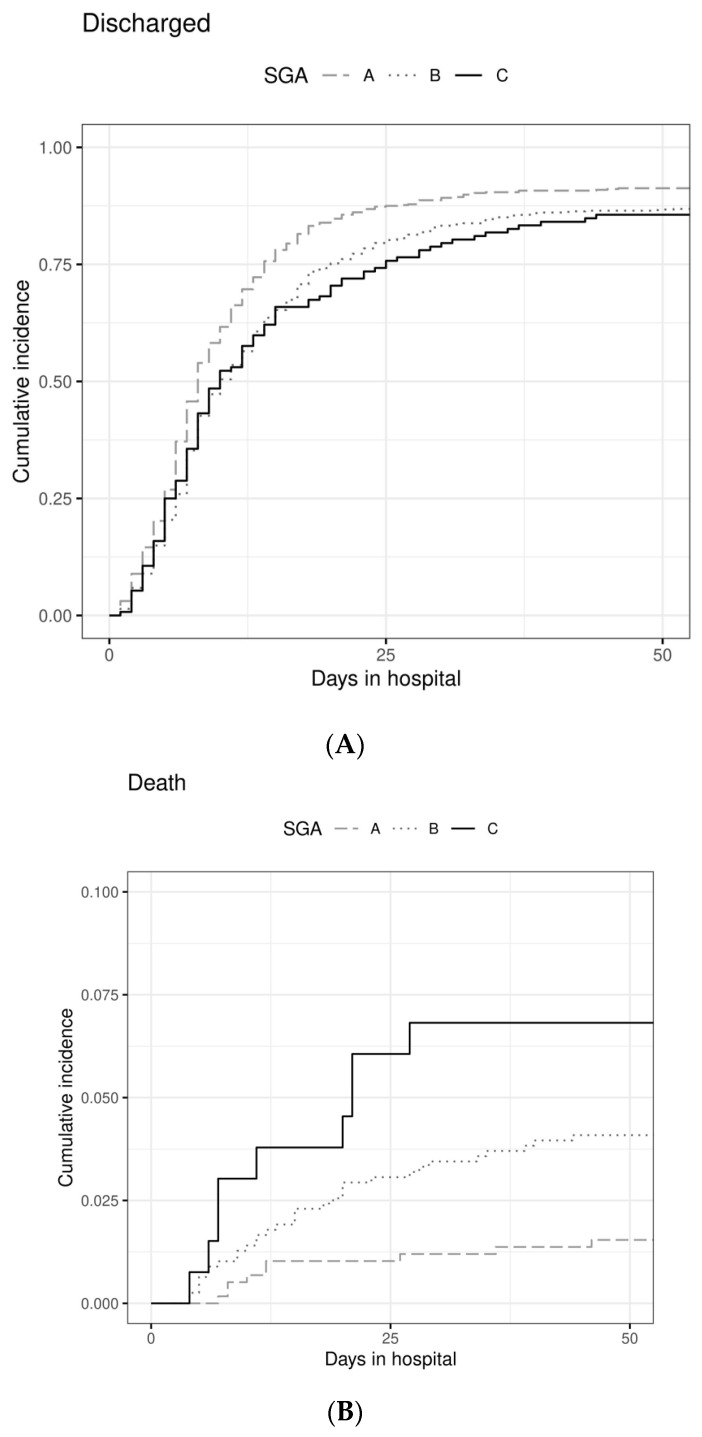
Cumulative incidence of discharge (**A**) and death (**B**) over 50 days after hospital admission according to SGA categories A (*n* = 584), B (*n* = 783), C (*n* = 131). SGA: subjective global assessment.

**Figure 3 nutrients-14-01780-f003:**
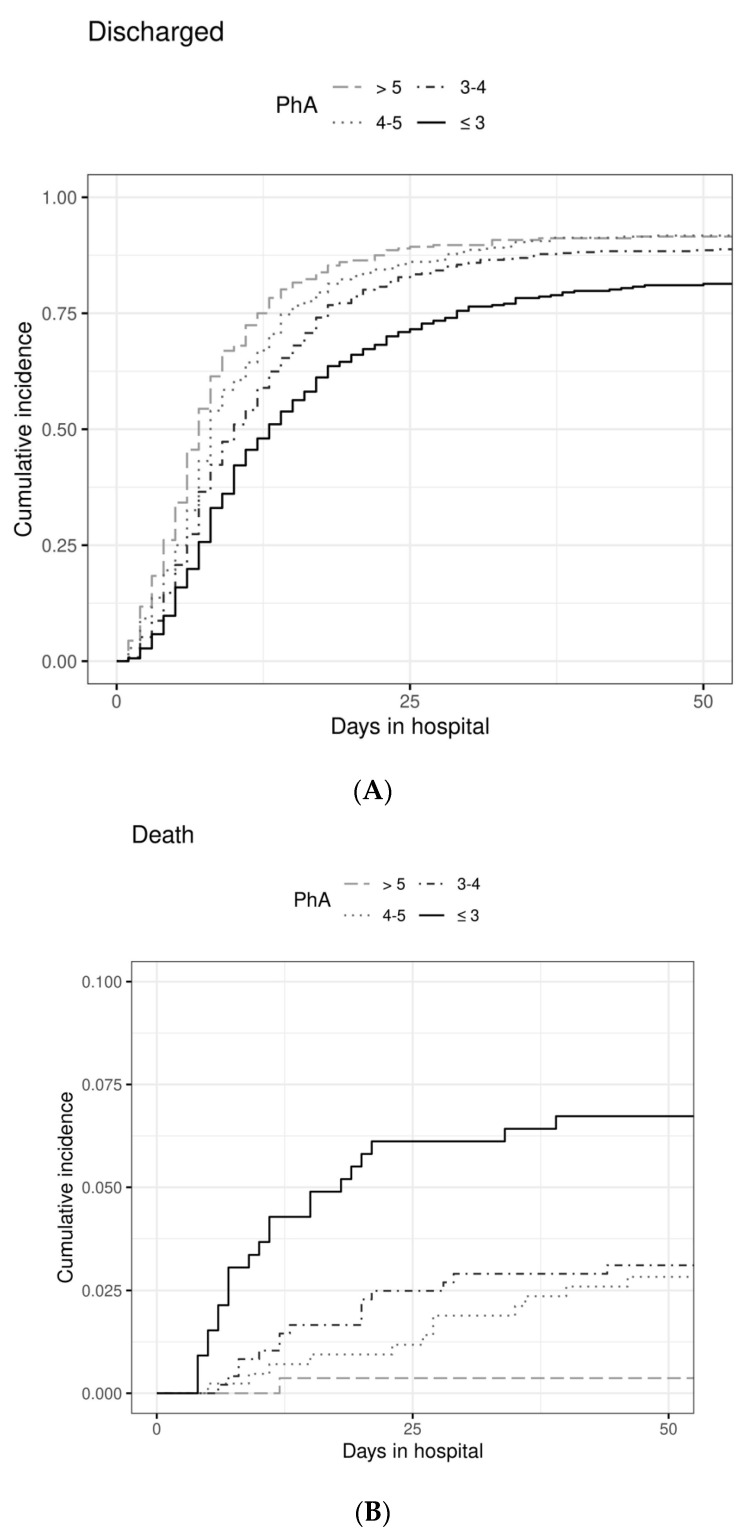
Cumulative incidence of discharge (**A**) and death (**B**) over 50 days after hospital admission according to PhA categories ≤ 3° (*n* = 327), 3–4° (*n* = 482), 4–5° (*n* = 424), > 5° (*n* = 272).

**Figure 4 nutrients-14-01780-f004:**
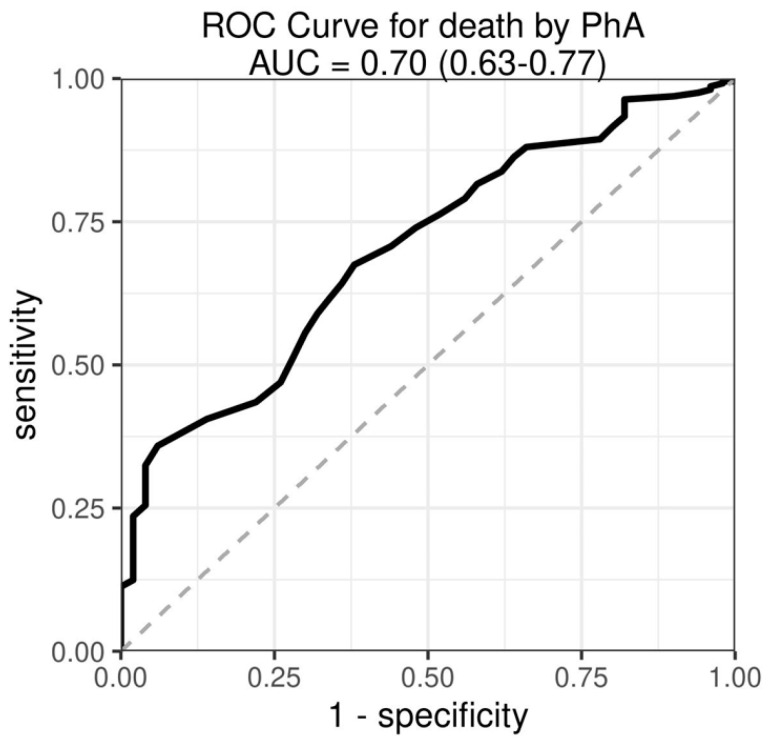
Receiver—operating—characteristic (ROC) curve for the prediction of in-hospital death by numerical PhA. AUC = area under the curve.

**Figure 5 nutrients-14-01780-f005:**
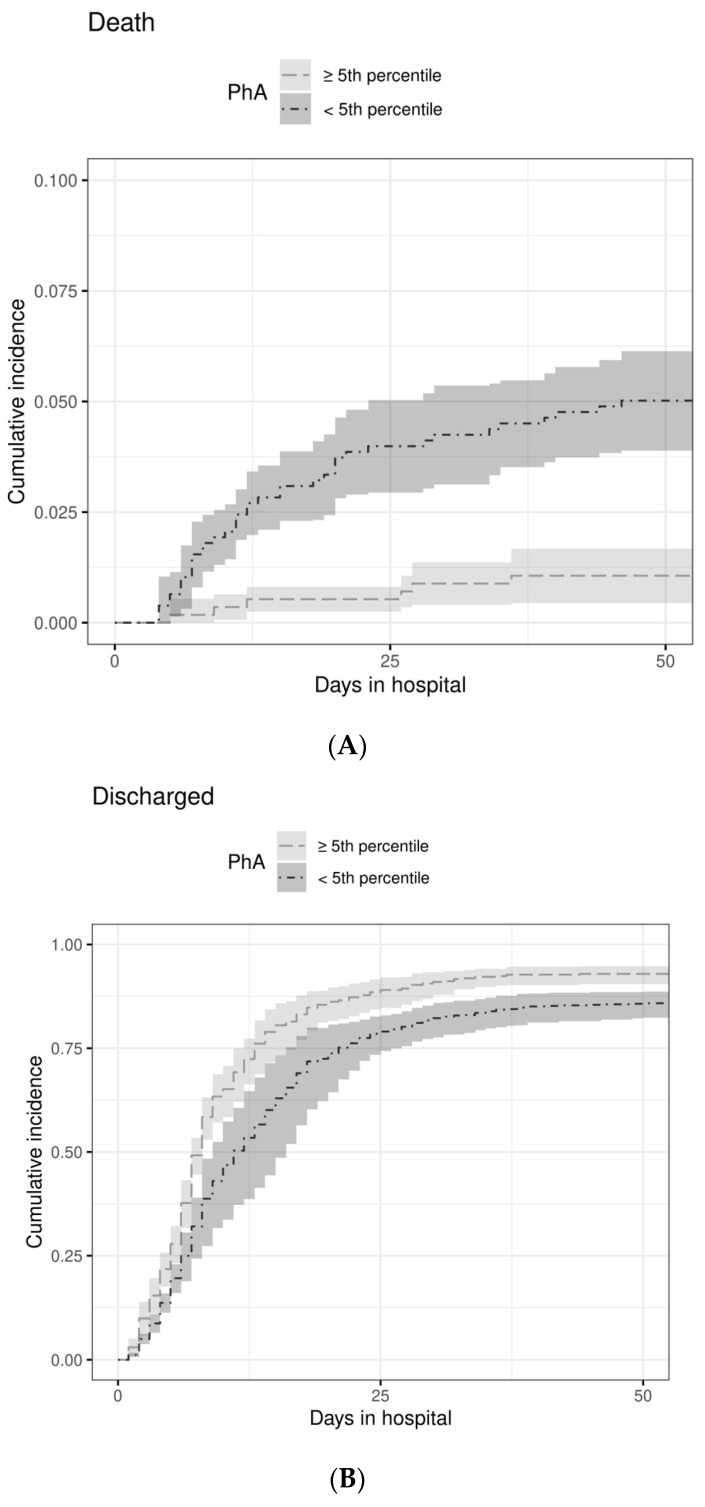
Cumulative incidence of death (**A**) and discharge (**B**) over 50 days after hospital admission according to PhA < 5th percentile (*n* = 777) or ≥ 5th percentile (*n* = 565) as classified according to age, sex, BMI-class, and population-specific reference values [26]. Areas in grey shade indicate 95%CI.

**Table 1 nutrients-14-01780-t001:** Demographic data of study cohort. * *n* = 1498.

*n*	1.505
age [years; median (IQR)]	76 (66–81)
female [*n* (%)]	782 (52%)
male [*n* (%)]	723 (48%)
height [mean ± SD]	167.8 ± 9.4 *
weight [mean ± SD]	69.4 ± 17.1 *
BMI [mean ± SD]	24.6 ± 5.5 *
weight gain [*n* (%)]	208 (14%)
weight unchanged [*n* (%)]	259 (18%)
weight loss (0–5%] [*n* (%)]	312 (22%)
weight loss (5–10%] [*n* (%)]	309 (21%)
weight loss > 10% [*n* (%)]	352 (24%)
weight loss missing	65 (4%)
SGA A [*n* (%)]	584 (39%)
SGA B [*n* (%)]	783 (52%)
SGA C [*n* (%)]	131 (9%)
SGA missing	6 (4%)

* *p* < 0.01. IQR: interquartile range; SGA: subjective global assessment.

**Table 2 nutrients-14-01780-t002:** Phase angle of study cohort.

*n*	1.505
phase angle percentile not defined due to BMI < 18.5	163 (11%)
phase angle [°; median (IQR)]	4.0 (3.2–4.7)
phase angle ≥ 5th perc [*n* (%)]	565 (38%)
phase angle < 5th perc [*n* (%)]	777 (52%)

**Table 3 nutrients-14-01780-t003:** ICD-10 classes of the study cohort.

*n*		1.505
ICD-10		*n* (%)
A–B	infections	78 (5%)
C–D50	neoplasms	356 (24%)
D50–D89	blood, blood-forming organs	72 (5%)
E	endocrine	35 (2%)
F	mental	5 (0%)
G	nervous system	39 (3%)
H00–59	eye	3 (0%)
H60–95	ear	8 (1%)
I	circulatory	176 (12%)
J	respiratory	110 (7%)
K	digestive	286 (19%)
L	skin	15 (1%)
M	musculoskeletal	60 (4%)
N	genitourinary	62 (4%)
P	pregnancy	2 (0%)
Q	malformations	1 (0%)
R	abnormal findings	65 (4%)
S-T	injury, poison	103 (7%)
Z	factors from health services	3 (0%)
missing		25 (2%)

**Table 4 nutrients-14-01780-t004:** Competing risk analysis using three proportional hazard models.

		Model I	Model II (Including PhA)	Model III (Including PhA and CRP)
Variable	*n* (%)	HR [95%CI]	HR [95%CI]	HR [95%CI]
SGA–A	584 (39%)	1.00	1.00	1.00
SGA–B/C	915 (61%)	0.74 [0.69,0.79] ***	0.94 [0.85,1.05]	0.98 [0.87,1.1]
Phase angle ≤ 3°	327 (22%)		0.47 [0.39,0.56] ***	0.59 [0.48,0.72] ***
Phase angle 3–4°	482 (32%)		0.66 [0.56,0.78] ***	0.8 [0.67,0.95] #
Phase angle 4–5°	424 (28%)		0.79 [0.69,0.9] **	0.83 [0.72,0.95] *
Phase angle > 5°	272 (18%)		1.00	1.00
Age <65	375 (25%)	0.85 [0.7,1.04]	0.75 [0.63,0.89] *	0.75 [0.63,0.89] *
Age 65–75	361 (24%)	1.00	1.00	1.00
Age 75–80	337 (22%)	0.92 [0.78,1.09]	0.9 [0.76,1.06]	0.9 [0.78,1.04]
Age > 80	432 (29%)	0.95 [0.84,1.07]	1 [0.88,1.14]	0.95 [0.83,1.08]
BMI < 18.5	162 (11%)	0.99 [0.87,1.12]	1.02 [0.91,1.14]	0.97 [0.84,1.11]
BMI 18.5–25	715 (48%)	1.00	1.00	1.00
BMI 25–30	379 (25%)	1.01 [0.87,1.17]	1.02 [0.88,1.19]	1.07 [0.92,1.25]
BMI > 30	241 (16%)	0.83 [0.72,0.94] *	0.85 [0.73,0.99]	0.89 [0.79,1.01]
CRP ≤ 10	480 (32%)			1.00
CRP 10–100	673 (45%)			0.75 [0.63,0.88] **
CRP > 100	158 (10%)			0.54 [0.44,0.65] ***
CRP—no value	194 (13%)			1.65 [1.27,2.16] **
Sex—male	723 (48%)	1.00	1.00	1.00
Sex—female	782 (52%)	1.05 [0.98,1.13]	1.09 [1.02,1.17]	0.99 [0.91,1.09]
ICD A–B: infections	79 (5%)	0.82 [0.67,0.99]	0.85 [0.71,1.02]	0.86 [0.69,1.07]
ICD C–D50: neoplasms	356 (24%)	0.62 [0.48,0.81] **	0.62 [0.47,0.82] **	0.61 [0.47,0.79] **
ICD D50–89: blood and blood-forming organs	72 (5%)	1.08 [0.75,1.56]	1.12 [0.77,1.62]	0.98 [0.66,1.45]
ICD E: endocrine	35 (2%)	1.38 [0.81,2.38]	1.31 [0.7,2.47]	1.2 [0.66,2.2]
ICD G: nervous system	39 (3%)	1.08 [0.77,1.51]	1.05 [0.75,1.47]	0.81 [0.54,1.21]
ICD I: circulatory	176 (12%)	0.58 [0.47,0.71] ***	0.57 [0.46,0.71] ***	0.52 [0.4,0.67] ***
ICD J: respiratory	110 (7%)	0.64 [0.49,0.83] **	0.62 [0.48,0.81] **	0.61 [0.49,0.77] ***
ICD 1 K: digestive	286 (19%)	1.00	1.00	1.00
ICD M: musculoskeletal	60 (4%)	0.61 [0.44,0.86] *	0.61 [0.43,0.86] *	0.56 [0.4,0.8] *
ICD N: genitourinary	62 (4%)	0.92 [0.61,1.39]	0.96 [0.63,1.45]	1.02 [0.76,1.35]
ICD: other	37 (2%)	0.69 [0.43,1.11]	0.61 [0.37,1]	0.52 [0.3,0.9]
ICD R: abnormal findings	65 (4%)	1.07 [0.65,1.78]	1.06 [0.63,1.81]	0.92 [0.54,1.57]
ICD S-T: injury, poison	103 (7%)	0.53 [0.43,0.64] ***	0.54 [0.45,0.64] ***	0.55 [0.47,0.64] ***

Model I: Age, sex, BMI, ICD-10 disease category, and SGA. Model II: PhA added to model I. Model III: CRP added to model II). All data are hazard ratios (HR) and 95% confidence intervals [in parentheses]. # *p* < 0.05, * *p* < 0.01, ** *p* < 0.001, *** *p* < 0.0001.

**Table 5 nutrients-14-01780-t005:** Logistic regression analysis for outcome in-hospital death using two models.

		Model I without PhA	Model II with Numerical PhA
Variable		OR [95%CI]	OR [95%CI]
SGA	A	1.00	1.00
	B/C	2.87 [1.38,5.94] *	1.16 [0.49,2.75]
PhA [°]	numerical		0.44 [0.32,0.61] ***
Age [years]	<65	2.12 [0.89,5.07]	3.68 [1.52,8.92] *
	65–75 y	1.00	1.00
	75–80 y	2.71 [1.08,6.8] #	2.79 [1.08,7.24] #
	>80 y	2.82 [1.03,7.73] #	2.27 [0.82,6.3]
BMI [kg/m^2^]	<18.5	0.84 [0.38,1.87]	0.64 [0.29,1.42]
	18.5–25	1.00	1.00
	25–30	0.92 [0.38,2.22]	0.8 [0.33,1.95]
	>30	1.01 [0.42,2.47]	0.82 [0.34,1.96]
Sex	M	1.00	1.00
	W	0.3 [0.16,0.58] **	0.26 [0.13,0.52] **
ICD-10	C-D50: neoplasms	2.28 [0.9,5.77]	2.49 [0.97,6.42]
	I: circulatory	1.03 [0.32,3.35]	1.14 [0.35,3.74]
	J: respiratory	1.41 [0.49,4.06]	1.64 [0.52,5.14]
	K: digestive	1.00	1.00
	other	0.37 [0.12,1.12]	0.37 [0.12,1.17]
	S-T: injury, poison	0.67 [0.15,3.03]	0.7 [0.16,3.04]

Model I: Age, sex, BMI, ICD-10 disease category, and SGA. Model II: Phase angle (PhA) added to model I as numerical value. All data are odds ratios (OR) and 95% confidence intervals [in parentheses]. # *p* < 0.05, * *p* < 0.01, ** *p* < 0.001, *** *p* < 0.0001.

## Data Availability

Data described in the manuscript will be made available upon request pending finalization of other secondary projects related to the trial.

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
