# Peer review of "Phase Angle Is a Stronger Predictor of Hospital Outcome than Subjective Global Assessment—Results from the Prospective Dessau Hospital Malnutrition Study"

_nutrients, 2022, doi:10.3390/nu14091780_

Round 1

Reviewer 1 Report

Phase Angle is a Stronger Predictor of Hospital Outcome than Subjective Global Assessment—Results from the Prospective Dessau Hospital Malnutrition Study

This is an interesting study comparing Phase Angle to the Subjective Global Assessment in regards to their ability to predict LOS in a large population of hospitalized patients. The data is well analysed and results are compelling suggesting that Phase Angle is superior to the Subjective Global Assessment in regards to outcome prediction.

I have only a few comments:

  • it would be interesting to also see a similar model that was used for LOS for the endpoint mortality. Is Phase Angle also superior in prediction of survival?
  • It would be interesting to see whether combination of Phase Angle and the Subjective Global Assessment is superior to either parameter by itself? An ROC of the combination of both vs both variabels may be of interest.
  • Recent paper suggested that handgrip strength was associated with outcome and also response to treatment (Kaegi Braun N, Am J Clin Nutr. 2021 Aug 2;114(2):731-740.  doi: 10.1093/ajcn/nqab042.). The authors may want to discuss this study in relation to their finding regarding PA results.

Author Response

Reviewer #1

It would be interesting to also see a similar model that was used for LOS for the endpoint mortality. Is Phase Angle also superior in prediction of survival?

         We thank the reviewer for addressing this issue. Due to the limited number of outcome events “death” the sample was not suitable for a multivariable analysis of competing risk. In order to address the impact of SGA and PhA on mortality we performed a logistic regression analysis in a simplified model. Results of this analysis confirm the superiority of PhA over SGA also for predicting mortality. We have included this analysis in the revised manuscript.

It would be interesting to see whether combination of Phase Angle and the Subjective Global Assessment is superior to either parameter by itself? An ROC of the combination of both vs both variables may be of interest.

         We discussed this suggestion of the reviewer and came to the conclusion that for the clinical practitioner a diagnostic algorithm of nutrition risk screening followed by determination of PhA together with the composite assessment tool SGA would give away the advantage of PhA providing a single objective, noninvasive measure in one step after prior screening.

         Moreover, we feel that combining a significant predictor with a non-significant predictor from a common multivariable model would not provide any additional benefit for clinical decision making.

         We appreciate the reviewer’s comment and added a ROC curve for the prediction of death by PhA alone.

Recent paper suggested that handgrip strength was associated with outcome and also response to treatment (Kaegi Braun N, Am J Clin Nutr. 2021 Aug 2;114(2):731-

  1. doi: 10.1093/ajcn/nqab042.). The authors may want to discuss this study in relation to their finding regarding PA results.

         We are grateful to the reviewer for this recommendation and have incorporated this study in the discussion.

Reviewer 2 Report

I have had the opportunity to review the paper entitled ‘Phase Angle is a Stronger Predictor of Hospital Outcome than Subjective Global Assessment—Results from the Prospective Dessau Hospital Malnutrition Study. The authors investigated the role of the bioelectrical phase angle in the prediction of outcome length of hospital stay in patients at risk for malnutrition. Overall, the study may have the potential to add meaningful information to the current body of literature. Detailed comments are listed below.

Abstract:

  • Report detailed info regarding the BIA technology used (e.g., foot-to-hand at a 50 kHz).
  • Define the CI considered.
  • Are all the acronymous used necessary? It is very hard to follow in this form. 

Introduction:

  • Overall, the introduction fails to provide a clear rationale as to why it is worthy to conduct a study using the bioelectrical phase angle as a marker of LOS. Additionally, several key studies from the past 10 years regarding phase angle and its associations with nutritional status, risk of fragility/sarcopenia, and other health-related outcomes are missing as well. I would encourage to authors to read more about the subject of phase angle and bioelectrical proprieties as these aspects need to be considered in the introduction to form a basis for this paper.
  • An important point regarding BIA is that it can be performed using a wide range of devices. However, no comparisons can be made between studies that estimate body composition with different technologies (e.g., foot-to-hand- or direct segmental in standing position) or sampling frequencies. I, therefore, suggest revising the introduction to make clear what phase angle represents.
  • According to the recent literature, it would be better to report phase angle as PhA instead of PA.

Methods

  • Authors should provide a more detailed description of the operation and validity of the impedance measurements. Is the NutriBox a phase-sensitive impedance device? What impedance variables are measured, and which are estimated? Describe the technical accuracy and precision of this device. Were the participants measured in the same time? (morning or evening?).
  • There is a basic need to describe the technical characteristics of this BIA device. Describe the probe and sound frequency (does it work in mono or multifrequency?). What is the calibration method to ensure validity (accuracy and precision) of the BIA measurements? What is the technical error of measurement in vivo? Provide readers with a concise description of what this device measures.
  • Why the BIVA approach was not applied to better classify the participants? 

Discussion

  • The discussion section is very descriptive and offers limited comparisons to previous research where PhA was identified as a valid biomarker of health-related outcomes. Similarly, how do practitioner benefit from that? Again, the discussion section fails to relate the findings to this particular application of interest.
  • It is important to consider that the phase angle is a dependent instrument and that the instrumental sensitivities are different. Therefore, no comparisons can be made between studies that measure PhA with different devices. The MDPI literature should be consulted as recently innovative studies have been proposed regarding BIA. 

References:

  • Please observe the MDPI guidelines

Author Response

Reviewer #2

Abstract:

Report detailed info regarding the BIA technology used (e.g., foot-to-hand at a 50 kHz).

         We included the appropriate information in the abstract.

Define the CI considered.

         In the abstract, we defined CI as 95% CI.

Are all the acronymous used necessary? It is very hard to follow in this form.

         We have made appropriate changes in the revised manuscript.

Introduction:

Overall, the introduction fails to provide a clear rationale as to why it is worthy to conduct a study using the bioelectrical phase angle as a marker of LOS. Additionally, several key studies from the past 10 years regarding phase angle and its associations with nutritional status, risk of fragility/sarcopenia, and other health-related outcomes are missing as well. I would encourage to authors to read more about the subject of phase angle and bioelectrical proprieties as these aspects need to be considered in the introduction to form a basis for this paper.

         We would like to thank the reviewer for this recommendation and have revised the section accordingly also including new references.

An important point regarding BIA is that it can be performed using a wide range of devices. However, no comparisons can be made between studies that estimate body composition with different technologies (e.g., foot-to-hand or direct segmental in standing position) or sampling frequencies. I, therefore, suggest revising the introduction to make clear what phase angle represents.

         We have revised the appropriate sections clarifying what phase angle represents and providing more details of the measurement protocol and the BIA device used.

According to the recent literature, it would be better to report phase angle as PhA instead of PA.

         We have revised the manuscript according to the reviewer’s recommendation.

Methods:

Authors should provide a more detailed description of the operation and validity of the impedance measurements.

Is the NutriBox a phase-sensitive impedance device?

         NutriBox is a phase sensitive device and we have included this information in the revised manuscript.

What impedance variables are measured, and which are estimated?

         Resistance and reactance are measured variables.

Describe the technical accuracy and precision of this device. Were the participants measured in the same time? (morning or evening?).

         All participants were measured at the same time (morning) and we have included appropriate information in the revised manuscript.

There is a basic need to describe the technical characteristics of this BIA device. Describe the probe and sound frequency (does it work in mono or multifrequency?). What is the calibration method to ensure validity (accuracy and precision) of the BIA measurements? What is the technical error of measurement in vivo?

         BIA measurements were made foot-to-hand in monofrequency mode using a 0.8 mA alternative current at 50 kHz. Precision of resistance measurements was ±1% and that of reactance was ±2.5%. We have included this information the revised manuscript.

Provide readers with a concise description of what this device measures.

         We have revised the methods section accordingly.

Why the BIVA approach was not applied to better classify the participants?

         We chose to use PhA because of the availability of population specific reference values from a large sample in Germany using the same BIA technology as in our study. No comparable population specific reference values are available for the BIVA approach.

Discussion:

The discussion section is very descriptive and offers limited comparisons to previous research where PhA was identified as a valid biomarker of health-related outcomes.

         We have revised the discussion section accordingly.

Similarly, how do practitioner benefit from that? Again, the discussion section fails to relate the findings to this particular application of interest.

         We are grateful to the reviewer for raising this question. To address this question we have extended the analysis of our data showing a significant association of PhA <5th percentile with prolonged hospital stay as well as mortality. Thus, the practitioner can benefit from clear diagnostic criteria.

It is important to consider that the phase angle is a dependent instrument and that the instrumental sensitivities are different. Therefore, no comparisons can be made between studies that measure PhA with different devices.

         We deepened the discussion of these issue in the revised manuscript.

The MDPI literature should be consulted as recently innovative studies have been proposed regarding BIA.

         We are grateful to the reviewer for this recommendation and have included further references.

References:

Please observe the MDPI guidelines

         We have revised the list of references accordingly.

Round 2

Reviewer 2 Report

Authors addressed all my comments